# Topoisomerase 1 cleavage complex enables pattern recognition and inflammation during senescence

Bo Zhao[1], Pingyu Liu[1], Takeshi Fukumoto[1], Timothy Nacarelli[1], Nail Fatkhutdinov [1], Shuai Wu[1], Jianhuang Lin [1], Katherine M. Aird[1], Hsin-Yao Tang [2], Qin Liu [3], David W. Speicher[2,3] & Rugang Zhang [1✉]

Cyclic cGMP-AMP synthase (cGAS) is a pattern recognition cytosolic DNA sensor that is essential for cellular senescence. cGAS promotes inflammatory senescence-associated secretory phenotype (SASP) through recognizing cytoplasmic chromatin during senescence. cGAS-mediated inflammation is essential for the antitumor effects of immune checkpoint blockade. However, the mechanism by which cGAS recognizes cytoplasmic chromatin is unknown. Here we show that topoisomerase 1-DNA covalent cleavage complex (TOP1cc) is both necessary and sufficient for cGAS-mediated cytoplasmic chromatin recognition and SASP during senescence. TOP1cc localizes to cytoplasmic chromatin and TOP1 interacts with cGAS to enhance the binding of cGAS to DNA. Retention of TOP1cc to cytoplasmic chromatin depends on its stabilization by the chromatin architecture protein HMGB2. Functionally, the HMGB2-TOP1cc-cGAS axis determines the response of orthotopically transplanted ex vivo therapy-induced senescent cells to immune checkpoint blockade in vivo. Together, these findings establish a HMGB2-TOP1cc-cGAS axis that enables cytoplasmic chromatin recognition and response to immune checkpoint blockade.

[1] Gene Expression and Regulation Program, The Wistar Institute, Philadelphia, PA 19104, USA. [2] Proteomics and Metabolomics Facility, The Wistar Institute, Philadelphia, PA 19104, USA. [3] Molecular and Cellular Oncogenesis Program, The Wistar Institute, Philadelphia, PA 19104, USA. ✉email: rzhang@wistar.org

Cellular senescence is a stress response associated with human diseases, such as cancer and aging[1]. Senescent cells are stably growth arrested[1]. Senescence is an important tumor suppression mechanism that limits the propagation of cells subjected to insults, such as activation of oncogenes or chemotherapeutics, known as oncogene-induced senescence or therapy-induced senescence, respectively[2]. A hallmark of senescent cells is the secretion of various cytokines, chemokines, growth factors, and proteases, collectively termed the senescence-associated secretory phenotype (SASP)[2]. The pattern recognition cGAS-STING pathway is important for the regulation of senescence and associated SASP[3–6]. cGAS is a cytosolic DNA sensor[7]. Notably, except for a certain length preference, cGAS senses double-stranded DNA (dsDNA) in a sequence nonspecific manner[7,8]. However, the molecular mechanisms by which cGAS senses dsDNA remain to be poorly understood. For example, the role of covalent DNA modification in dsDNA sensing by cGAS has not been explored.

cGAS regulates de novo transcriptional immune response through downstream STING to activate type I interferons (IFN) through IRF3, and proinflammatory cytokines and chemokines through NFκB[9]. The activation of cGAS in senescent cells is due to the induction of cytoplasmic chromatin fragments (CCF) caused by nuclear membrane blebbing[3,10]. In addition, cell-cycle progression through mitosis following DNA double-strand breaks has been shown to cause the formation of micronuclei[11,12]. Thus, it is possible that some CCF are caused by micronuclei formation during the mitosis prior to senescence-associated cell-cycle exit and in particular in senescent cells induced by DNA-damaging agents. CCF contain genomic DNA and are positive for DNA damage marker γH2AX[3,10]. However, the molecular basis by which cGAS recognizes CCF to promote inflammation during senescence is unknown.

Members of the high-mobility group proteins (HMG) are nonhistone proteins that bind DNA to regulate chromatin architecture[13]. For example, high-mobility group box 2 (HMGB2) bends DNA without sequence specificity[14]. Notably, HMGB2 orchestrates gene expression reprogramming to promote SASP during senescence[15]. However, the role of HMGB2 in CCF regulation has never been explored. In addition, HMGB proteins are implicated in sensing nucleic acids and stimulating long DNA sensing by cGAS[16,17]. However, the mechanism by which HMGB2 regulates sensing of nucleic acids by cGAS remains to be explored.

Topoisomerase 1 (TOP1) is responsible for relaxing higher order topological DNA structures during DNA replication and gene transcription[18]. TOP1 forms a stable protein–DNA cleavage complex (TOP1cc) through its enzymatic activity, and TOP1 becomes covalently bound to the catalytically generated DNA strand break[18]. Trapped or persistent TOP1cc induced by TOP1 inhibitors such as camptothecin are harmful to normal cellular function because they block both DNA and RNA polymerases[18]. However, the role of TOP1cc in senescence has never been explored.

cGAS is essential for the antitumor effect of immune checkpoint blockades such as anti-PD-L1 antibody[19]. Here we report that TOP1cc plays a critical role in mediating recognition of CCF by cGAS, and the associated SASP during senescence. Mechanistically, HMGB2 stabilizes TOP1cc to enhance the binding of cGAS to dsDNA. Indeed, the HMGB2-TOP1cc-cGAS axis determines the response of orthotopically transplanted ex vivo therapy-induced senescent cells to immune checkpoint blockade in vivo.

## Results

### HMGB2 is required for cGAS' localization into CCF during senescence. Since HMGB2 positively regulates SASP[15] and facilitates cytosolic nucleic acid sensing[16], we examined whether HMGB2 localized to the CCF during senescence. HMGB2 co-localized with γH2AX in the CCF in senescent OVCAR3 ovarian cancer cells induced by either cisplatin or etoposide (Supplementary Fig. 1a–d). Indeed, HMGB2 co-localized with cGAS and γH2AX in the CCF in therapy-induced senescent OVCAR3 cells (Fig. 1a). We next determined the effects of HMGB2 loss on CCF, and recognition of CCF by cGAS. We generated two HMGB2 knockout OVCAR3 clones (Fig. 1b). HMGB2 knockout did not affect CCF formation as examined by γH2AX's localization to CCF (Fig. 1c–f; Supplementary Fig. 1e). However, HMGB2 knockout significantly decreased the localization of cGAS into the CCF (Fig. 1c–f). Consistent with previous reports that HMGB2 knockdown selectively suppresses SASP but does not affect senescence-associated growth arrest[15], HMGB2 knockout did not affect other markers of senescence such as SA-β-Gal activity and downregulation of proliferation marker cyclin A (Supplementary Fig. 1a–c). This indicates that the observed changes in cGAS localization were not a consequence of senescence suppression by HMGB2 knockout. Similar findings were also made in oncogenic H-RAS[G12V] or etoposide-induced senescent primary embryonic lung fibroblast IMR90 cells, with or without shRNA-mediated HMGB2 knockdown (Fig. 1g–h; Supplementary Fig. 2). Notably, HMGB2 knockout or knockdown did not decrease cGAS expression (Fig. 1b; Supplementary Fig. 2a), suggesting that the observed loss of cGAS' localization into CCF was not due to a decrease in cGAS protein expression. Consistent with a significant decrease in cGAS localization to CCF, HMGB2 knockdown significantly decreased the levels of secreted SASP factors as determined by an antibody-based array (Fig. 1i). Likewise, mRNA expression of SASP genes was also suppressed by HMGB2 knockdown (Supplementary Fig. 2h). Together, we conclude that HMGB2 localizes to CCF and is required for cGAS' localization to CCF.

### cGAS activation requires TOP1cc during senescence. We next determined the mechanism by which HMGB2 regulates cGAS' localization into CCF during senescence. Toward this goal, we developed a protocol to purify CCF from senescent cells (Supplementary Fig. 3a–c). Transfection of the purified CCF from etoposide-induced senescent IMR90 cells upregulated the expression of SASP genes in naive IMR90 cells, validating the protocol we developed (Supplementary Fig. 3d, e). We next performed stable isotope labeling with amino acids in cell culture (SILAC) by labeling etoposide-induced senescent IMR90 cells with or without inducible HMGB2 knockdown with light or heavy isotopes, respectively (Supplementary Fig. 3f). We isolated the CCF from these cells and performed liquid chromatography tandem mass spectrometry (LC-MS) analysis to identify proteins that are differentially localized to CCF in senescent cells, with versus without HMGB2 knockdown. We focused our analysis on proteins that are implicated in the nucleosome and chromosome-related functionality, given that CCF formed by nuclear membrane blebbing are positive for chromatin markers[3,10]. The analysis revealed that topoisomerase 1 (TOP1) was among the top differentially proteins in CCF isolated from senescent cells, with or without HMGB2 knockdown. TOP1 levels in CCF were increased by HMGB2 knockdown compared with control senescent cells (Supplementary Fig. 3g). Notably, TOP1 forms TOP1cc without strict DNA sequence preference[18]. Thus, TOP1 exists in two forms: free TOP1 and TOP1cc covalently bound to dsDNA[18]. Notably, inhibition of TOP1 activity by camptothecin (CPT) leads to trapping of TOP1cc on DNA, and thus increases TOP1cc levels[18].

We first validated the unbiased LC-MS results by showing that TOP1 localized to CCF and co-localized with γH2AX in both

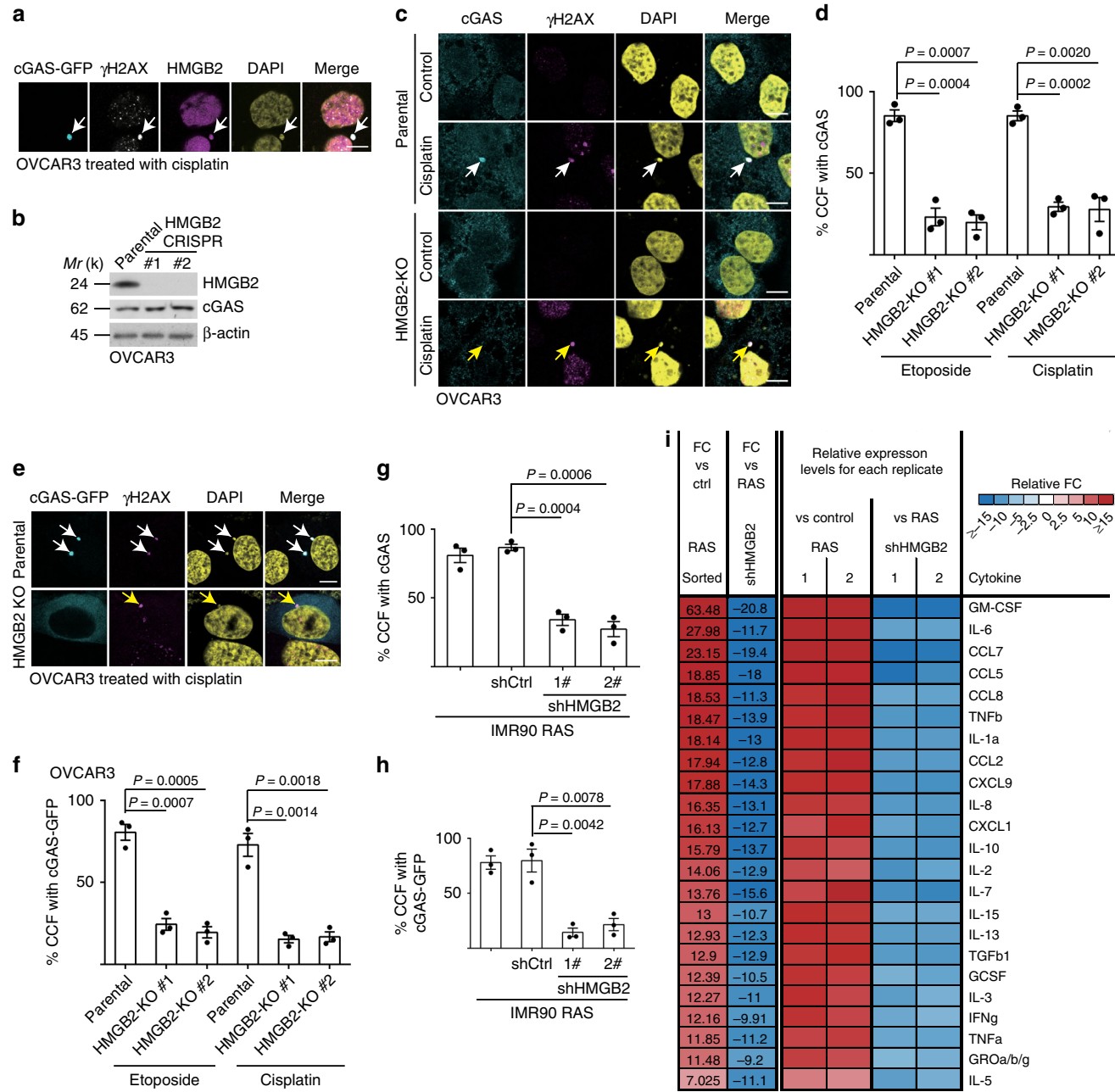

**Fig. 1 HMGB2 is required for cGAS' localization into CCF during senescence. a** OVCAR3 cells expressing GFP-tagged cGAS were induced to senesce by cisplatin, and imaged under a confocal microscope. cGAS-GFP, γH2AX, and HMGB2 co-localized CCF are indicated by arrows. **b** Expression of HMGB2, cGAS, and a loading control β-actin in the indicated parental and two independent HMGB2 knockout OVCAR3 clones determined by immunoblot. **c**, **d** Representative images (**c**) and quantification (**d**) of endogenous cGAS localization into γH2AX-positive CCF in senescent parental and HMGB2 knockout OVCAR3 cells with the indicated treatment. Arrows point to CCF. **e**, **f** Representative images (**e**) and quantification (**f**) of cGAS-GFP localization into γH2AX-positive CCF in senescent parental and HMGB2 knockout OVCAR3 cells with the indicated treatment. Arrows point to CCF. **g**, **h** Quantification of endogenous cGAS (**g**) or cGAS-GFP (**h**) localization into CCF in senescent primary IMR90 cells induced by oncogenic RAS, with or without HMGB2 knockdown. **i** The secretion of soluble factors under the indicated conditions were detected by antibody arrays. The heatmap indicates the fold change (FC) in comparison with the control (Ctrl) or RAS-induced senescent IMR90 cells. Relative expression levels per replicate and average fold change differences are shown. Data represent mean ± s.e.m. n = 3 biologically independent experiments unless otherwise stated. Scale bar = 10 μm. *P*-values were calculated using a two-tailed *t* test. Source data are provided as a Source Data file.

senescent IMR90 and OVCAR3 cells (Supplementary Fig. 4a, b). We further validated that TOP1 levels in CCF were increased by HMGB2 knockdown in senescent IMR90 cells (Fig. 2a) and by HMGB2 knockout in senescent OVCAR3 cells (Supplementary Fig. 4c). TOP1 levels in CCF were increased by HMGB2 inhibition that suppresses SASP, suggesting that TOP1 may negatively regulate SASP. However, knockdown of TOP1 significantly suppressed the

expression of SASP genes (Supplementary Fig. 4d, e), suggesting that the presence of TOP1 in CCF positively regulates SASP. Thus, although TOP1 levels in CCF were increased in HMGB2-inhibited senescent cells, TOP1 may positively regulate SASP. Therefore, we instead examined the localization of TOP1cc in CCF in senescent cells with or without HMGB2 inhibition. Indeed, TOP1cc localized to CCF and co-localized with γH2AX in CCF (Fig. 2b, c). However,

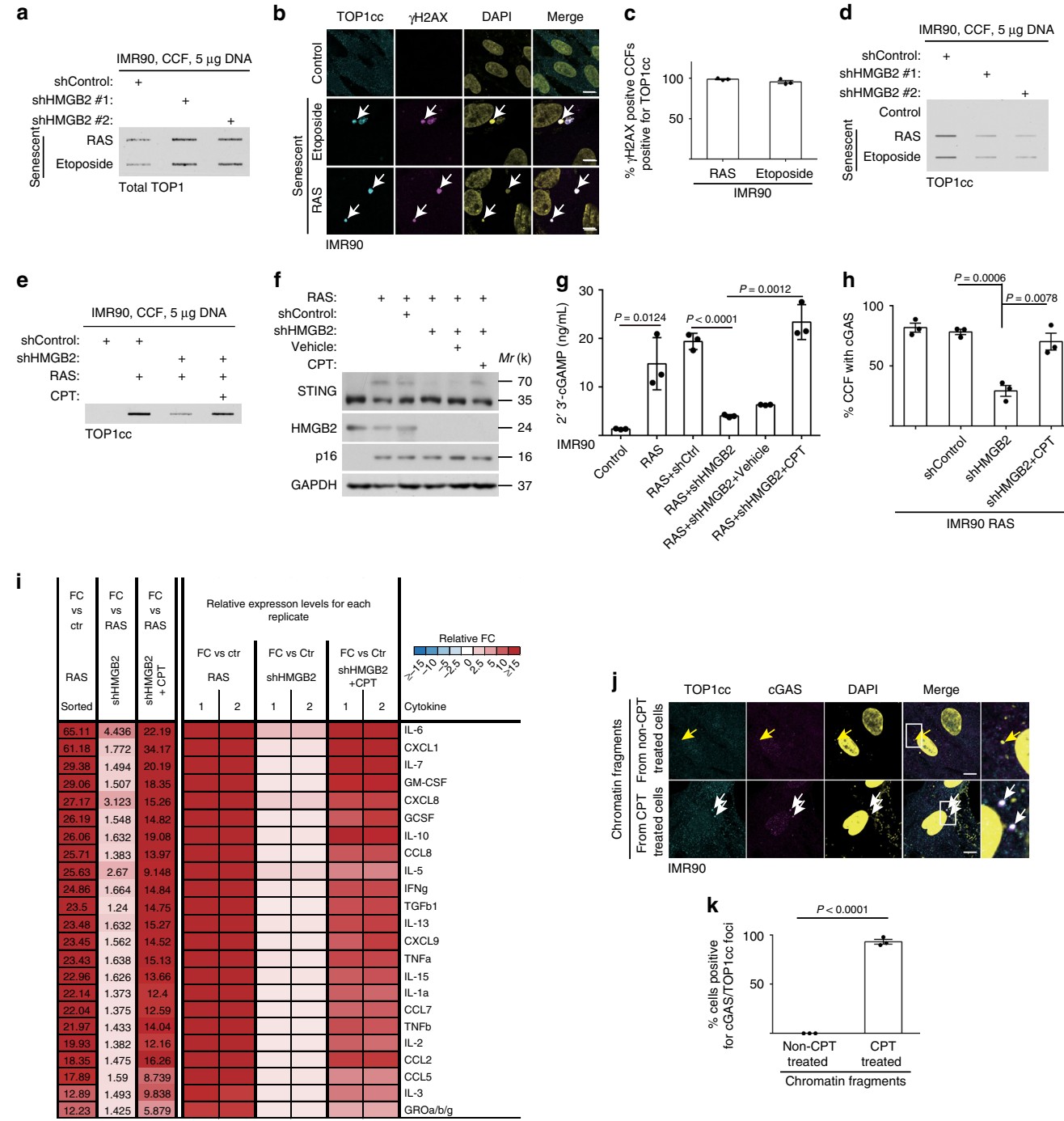

**Fig. 2 cGAS activation requires TOP1cc during senescence. a** Slot blot analysis of total TOP1 proteins in CCF purified from senescent IMR90 cells induced by the indicated treatment, with or without HMGB2 knockdown. **b, c** IMR90 cells induced to senesce by etoposide or oncogenic RAS were stained for TOP1cc and γH2AX (**b**), and percentages of γH2AX-positive CCF positive for TOP1cc were quantified (**c**). CCF are indicated by arrows. **d** Slot blot analysis of TOP1cc levels in CCF purified from senescent IMR90 cells induced by the indicated treatment with or without HMGB2 knockdown. **e** Slot blot analysis of TOP1cc levels in CCF purified from the indicated IMR90 cells treated with or without TOP1 inhibitor Camptothecin (CPT). **f, g** Examination of STING dimerization (**f**) or 2′3′-cGAMP levels (**g**) in the indicated cells. **h** Quantification of cGAS localization into CCF in the indicated IMR90 cells. **i** The secretion of soluble factors under the indicated conditions were detected by antibody arrays. The heatmap indicates the fold change (FC) in comparison with the control (Ctr) or RAS condition. Relative expression levels per replicate and average fold change differences are shown. **j, k** IMR90 cells were transfected with chromatin fragments isolated from IMR90 cells with or without CPT treatment. Benzonase was used to digest chromatin into fragments. Representative images (**j**) and quantification (**k**) of cGAS and TOP1cc co-localization in the transfected IMR90 cells. Arrows point to cGAS foci induced by the transfected chromatin fragments without or with TOP1cc. Data represent mean ± s.e.m. $n = 3$ biologically independent experiments unless otherwise stated. Scale bar = 10 μm. *P*-values were calculated using a two-tailed *t* test. Source data are provided as a Source Data file.

in contrast to an increased level of TOP1 in CCF, TOP1cc levels in CCF were decreased by HMGB2 knockdown or knockout (Fig. 2d; Supplementary Fig. 5a), which is consistent with the finding that HMGB2 loss suppresses CCF-mediated SASP (Fig. 1).

Since our results suggest that TOP1cc promotes SASP, we next directly examined whether induction of TOP1cc by CPT is sufficient to rescue the suppression of SASP induced by HMGB2 inhibition. Notably, CPT treatment restored the TOP1cc levels in the CCF isolated from HMGB2 knockdown or knockout senescent cells (Fig. 2e; Supplementary Fig. 5b). In addition, CPT treatment rescued the suppression of STING dimerization and downregulation of 2'3'-cGAMP levels induced by HMGB2 knockdown (Fig. 2f, g), which correlated with a rescue of the localization of cGAS and TOP1cc into CCF in HMGB2 knockout cells (Fig. 2h; Supplementary Fig. 5c–f) and the restoration of the secretion of SASP factors as determined by an antibody array (Fig. 2i). Similar rescue was also observed for expression of cGAS-STING regulated type I IFN target gene ISG15 (Supplementary Fig. 5g). We next determined whether TOP1cc is sufficient to drive cGAS localization into CCF and upregulate SASP genes. We isolated genomic chromatin fragments from IMR90 cells treated with two doses of CPT that induced TOP1cc in a dose-dependent manner (Supplementary Fig. 5h). Transfection of the isolated TOP1cc-containing genomic chromatin fragments was indeed sufficient to induce SASP gene expression in a dose-dependent manner (Supplementary Fig. 5i, j). Notably, TOP1cc-containing genomic chromatin fragments induced the co-localization of TOP1cc and cGAS (Fig. 2j, k). The observed SASP induction by TOP1cc-containing genomic chromatin fragments was cGAS dependent, because cGAS knockdown abrogated the observed SASP induction (Supplementary Fig. 5i, j). Together, these results support that TOP1cc functions downstream of HMGB2 and upstream of cGAS.

**TOP1cc enhances dsDNA recognition by cGAS.** Since HMGB2 positively regulates TOP1cc and HMGB2 inhibition decreases TOP1cc, we examined time-course kinetics of TOP1cc induction and stabilization in CPT-treated IMR90 cells with or without HMGB2 knockdown. Notably, HMGB2 knockdown did not affect the kinetics of TOP1cc formation (Fig. 3a, b). In contrast, HMGB2 knockdown significantly decreased TOP1cc levels once the cells were released from CPT treatment (Fig. 3c, d). These results support that HMGB2 stabilizes TOP1cc.

Since SASP induction by TOP1cc is cGAS dependent, TOP1cc is a TOP1 covalently modified dsDNA complex[18], and cGAS binds to dsDNA[7], we examined whether TOP1 interacts with cGAS by co-immunoprecipitation analysis. Indeed, TOP1 interacted with cGAS, and there was an increase in their interaction in senescent compared with control cells (Fig. 4a, b). Notably, TOP1 directly interacts with cGAS in a GST pull-down assay (Supplementary Fig. 6a). Interestingly, the interaction between cGAS and TOP1 is HMGB2 dependent, because HMGB2 knockdown abrogated the interaction in DNA free co-IP lysates (Fig. 4b). This result suggests that cGAS interacts with TOP1cc, the DNA bound form of TOP1, because HMGB2 stabilizes TOP1cc which may explain the lack of interaction between cGAS and TOP1 in HMGB2 knockdown cells. Indeed, addition of a synthesized 45 bp interferon stimulatory dsDNA (ISD)₂[20] into to the lysates of HMGB2 knockdown cells to allow for TOP1cc formation significantly rescued the interaction between TOP1 and cGAS in HMGB2 knockdown senescent cells (Fig. 4b). Notably, DNase I treatment significantly reduced the intensity of DAPI-stained DNA in CCF (Supplementary Fig. 6b, c). However, the localization of TOP1 into the CCF was not affected by DNase I treatment (Supplementary Fig. 6b, c). This result suggests that

TOP1 can localize into CCF independent of DNA, which is consistent with our findings that HMGB2 knockdown reduced the TOP1cc levels while increased TOP1 levels in CCF (Fig. 2). Together, these results show that TOP1cc interacts with cGAS and HMGB2 regulates the interaction through controlling TOP1cc stability.

We next sought to directly determine the effects of TOP1cc on the DNA binding affinity of cGAS. Electrophoretic mobility shift assay (EMSA) showed high-molecular-weight cGAS-bound (ISD)₂ dsDNA complex in a dose-dependent manner (Supplementary Fig. 6d, e). In addition, EMSA showed that compared with wild-type TOP1, a point mutant TOP1 Y723F, that is defective in DNA binding and thus cannot form TOP1cc, was severely impaired in its ability to shift the free (ISD)₂ dsDNA (Fig. 4c). Significantly, wild-type TOP1, but not the TOP1 Y723F mutant, markedly enhanced the dsDNA-binding affinity of cGAS (Fig. 4d; lane 7 vs. 8). Together, these results support that TOP1cc formed by DNA binding wild-type TOP1 enhances dsDNA recognition by cGAS (Fig. 4e).

**HMGB2-TOP1cc-cGAS determines response to checkpoint blockade.** There is evidence to support that cGAS and its mediated expression of immune modulatory molecules such as SASP factors are essential for the antitumor effect of immune checkpoint blockade such as anti-PD-L1 antibody treatment[19]. To examine the relevance of the HMGB2-TOP1cc-cGAS pathway in immune checkpoint blockade treatment, we utilized an immune competent syngeneic ovarian cancer ID8-Defb29/Vegf-a mouse model[21,22]. Notably, the HMGB2-cGAS-TOP1cc axis is conserved in cisplatin-induced senescent ID8-Defb29/Vegf-a cells and cisplatin-induced senescence in ~100% of the treated cells (Supplementary Fig. 7). To examine the effects of HMGB2 loss during senescence on the response to the anti-PD-L1 antibody treatment, we treated ID8-Defb29/Vegf-a cells with cisplatin to induce senescence ex vivo with or without inducible HMGB2 knockdown as previously shown for radiation-induced cGAS-mediated inflammatory response[11]. Then, we orthotopically transplanted the senescent cells into C57BL/6 mice by i.p. injection. Two weeks after transplantation, we randomized mice into different treatment groups. Compared with control tumors treated with anti-PD-L1 antibody, HMGB2 knockdown significantly abrogated the response to anti-PD-L1 antibody treatment (Fig. 5a–c). This correlated with suppression of the expression of SASP genes both in vitro and in vivo in the sorted orthotopically transplanted tumor cells (Fig. 5d; Supplementary Fig. 8a). Since HMGB2 is required for cGAS-dependent activation of SASP genes, these results are consistent with the literature that cGAS and its regulated immune modulating molecules such as SASP are essential for the antitumor effect of immune checkpoint blockade[19]. Since CPT treatment rescues recognition of CCF by cGAS and SASP when HMGB2 is inhibited (Fig. 2e–i), we treated the HMGB2 knockdown tumors with CPT to determine whether CPT treatment is sufficient to restore the anti-PD-L1 treatment response in these tumors. Indeed, CPT treatment significantly restored the anti-PD-L1 response (Fig. 5b, c). Consistently, compared with control tumors treated with anti-PD-L1, HMGB2 knockdown erased the survival advantage improved by anti-PD-L1 antibody treatment (Fig. 5e). Notably, CPT treatment rescued the survival of mice bearing HMGB2 knockdown tumors to a degree that is comparable with mice bearing control tumors treated with an anti-PD-L1 antibody (Fig. 5e). However, CPT treatment did not affect the body weight of the treated mice (Supplementary Fig. 8b), suggesting that CPT did not exhibit toxicity in anti-PD-L1 antibody-treated group. Consistent with a requirement for T-cell responses in the observed tumor suppressive effects

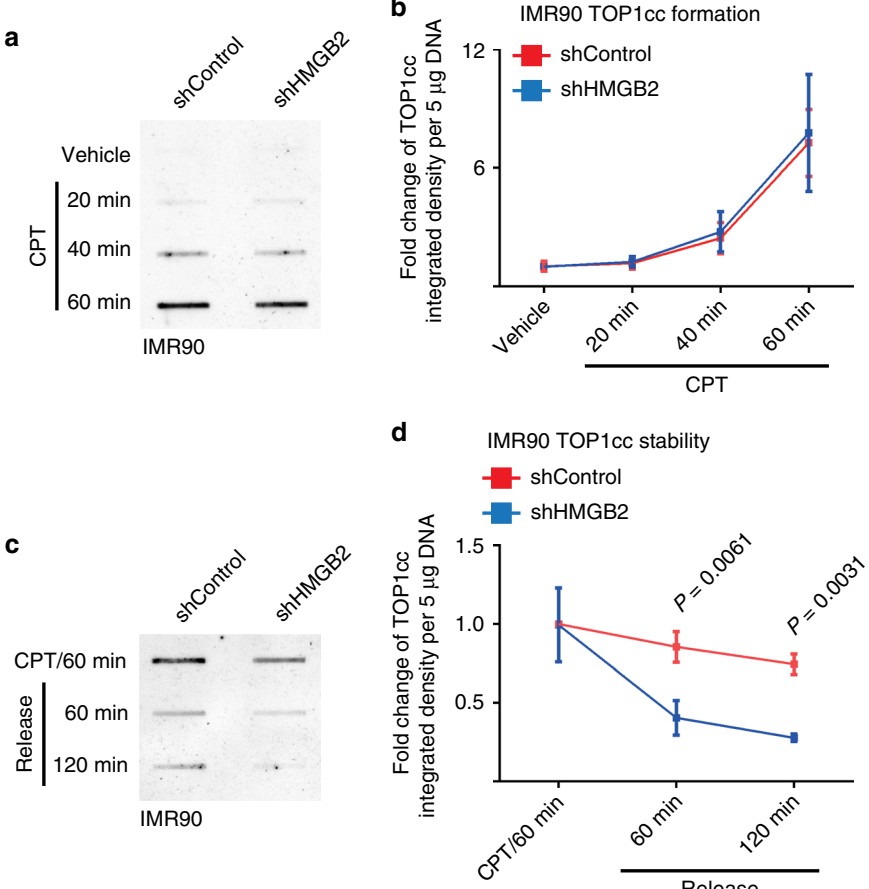

**Fig. 3 HMGB2 stabilizes TOP1cc. a, b** IMR90 cells with or without HMGB2 knockdown treated with CPT were analyzed for TOP1cc levels at the indicated time points by slot blot (**a**). Fold changes in TOP1cc levels at the indicated time points were quantified (n = 3) (**b**). **c, d** IMR90 cells with or without HMGB2 knockdown treated with CPT for 60 min and released from the treatment for 60 or 120 min were analyzed for TOP1cc levels by slot blot (**c**). Fold changes in TOP1cc levels at the indicated release time points were quantified (n = 3) (**d**). Data represent mean ± s.e.m. n = 3 biologically independent experiments. P-values were calculated using a two-tailed t test. Source data are provided as a Source Data file.

by anti-PD-L1 blockade, both activated CD69[+]/CD8[+] and IFNγ[+]/CD8[+] T cells correlated with changes in survival in the different treatment groups (Fig. 5f, g; Supplementary Fig. 8c). Notably, the activated CD69[+]/CD4[+] or Granzyme B[+]/CD8 T cells were not changed among the different treatment groups (Supplementary Fig. 8d, e). Together, we conclude that the status of the HMGB2-TOP1cc-cGAS axis determines the response to immune checkpoint blockade.

## Discussion

Consistent with previous reports[15], HMGB2 knockdown suppresses the growth of the tumor cells (Fig. 5c). However, HMGB2 expression is critical for response to checkpoint blockade in the context of therapy-induced senescence. This is due to its role in mediating SASP that is important for checkpoint blockade response. Thus, the role of HMGB2 in therapy response is context dependent. In addition, HMGB2 knockdown suppressed SASP and reduced the tumor growth in vivo (Fig. 5b, c), which is consistent with the previous notion that SASP promotes tumor growth in a context-dependent manner[23]. HMGB2 knockdown or knockout increased TOP1 levels in CCF (Fig. 4e). However, this was not sufficient to compensate for the decrease in TOP1cc levels in CCF. Thus, the lack of TOP1cc due to its destabilization contributes to suppression of SASP by HMGB2 inhibition. This also explains the increase in TOP1 and a decrease in TOP1cc

levels in CCF of HMGB2 knockdown or knockout senescent cells (Fig. 4e). Thus, our findings identified a critical component in the cGAS-mediated inflammation response by providing a molecular mechanism through which cytoplasmic chromatin is recognized by cGAS.

Here we identified the TOP1cc, a TOP1 covalently modified DNA complex, as a critical mediator of the recognition of CCF by cGAS through direct interaction between TOP1 and cGAS in a dsDNA-dependent manner during senescence. In addition, we showed that HMGB2 functions upstream of the TOP1cc-cGAS axis by stabilizing TOP1cc. Thus, our studies provided additional mechanistic insights into how HMGB proteins boost cytosolic nucleic acid sensing[16]. Finally, we show that the HMGB2-TOP1cc-cGAS axis functionally regulates SASP and the response to immune checkpoint blockade. These findings suggest that clinically applicable TOP1 inhibitors such as CPT may serve as a sensitizer to immune checkpoint blockade to target therapy-induced senescent cells. Notably, TOP1 inhibitors increase the sensitivity of patient-derived melanoma cell lines to T-cell-mediated cytotoxicity[24,25]. This is consistent with our findings that TOP1 inhibitors-induced TOP1cc boosts cGAS-mediated inflammation and the associated immune checkpoint blockade treatment. These findings suggest possibilities to modulate the HMGB2-TOP1cc-cGAS axis during senescence-inducing therapy through boosting host immune system.

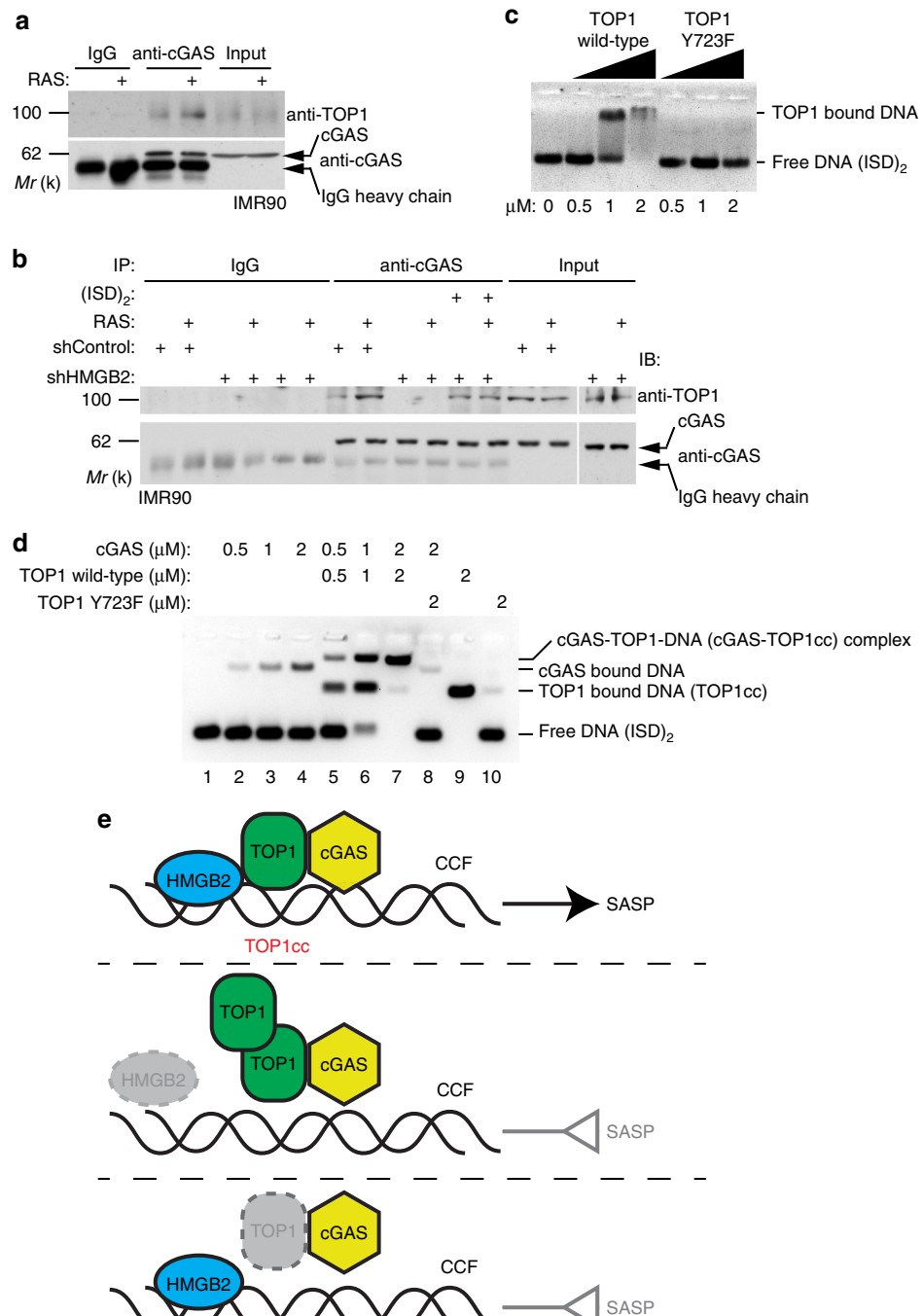

**Fig. 4 TOP1cc enhances dsDNA recognition by cGAS. a** Co-immunoprecipitation analysis of cGAS and TOP1 in control and RAS-induced senescent IMR90 cells. An isotype matched IgG was used as a negative control. **b** Co-Immunoprecipitation analysis of cGAS and TOP1 in control and RAS-induced senescent IMR90 cells with or without HMGB2 knockdown. Note that addition of (ISD)$_2$ dsDNA into the supernatant rescues the interaction between cGAS and TOP1 in HMGB2 knockdown senescent IMR90 cells. **c** Electrophoretic mobility shift analysis of TOP1 wild-type and TOP1 Y723F mutant proteins. **d** Electrophoretic mobility shift analysis shows that wild-type TOP1, but not the mutant TOP1 Y723, enhances the binding of cGAS to dsDNA. **e** A proposed model as described in the text. Data represent mean ± s.e.m. $P$-values were calculated using a two-tailed $t$ test. Source data are provided as a Source Data file.

## Methods

**Cells and culture conditions**. IMR90 human diploid fibroblasts were cultured according to American Type Culture Collection (ATCC) under low oxygen tension (2%) in DMEM (4.5 g per liter glucose) supplemented with 10% fetal bovine serum (FBS), L-glutamine (Thermo Fisher, Cat. No. 25030081), sodium pyruvate (Thermo Fisher, Cat. No. 11360070), nonessential amino acid (Thermo Fisher, Cat. No. 11140-050), sodium bicarbonate (Gibco, Cat. No. 25080094) and 1% penicillin/streptomycin (Corning, Cat. No. 30-002-CI). All experiments were performed on IMR90 fibroblasts cultured between the population doubling of 25 and 35. Human

ovarian cancer cell line OVCAR3 obtained from ATCC and mouse ovarian cancer cell line ID8-*Defb29/Vegf-a* gifted by Dr. Jose R. Conejo-Garcia were cultured in RPMI 1640 supplemented with 10% FBS and 1% penicillin/streptomycin. All the cells lines are authenticated at The Wistar Institute's Genomics Facility using short tandem repeat DNA profiling. Regular mycoplasma testing was performed using the LookOut Mycoplasma PCR detection (Sigma, Cat. No. MP0035).

**Reagents, plasmids, and antibodies**. Etoposide was purchased from Sigma (Cat. No. E1383). Cisplatin was purchased from Selleck (Cat. No. S1166). Doxycycline

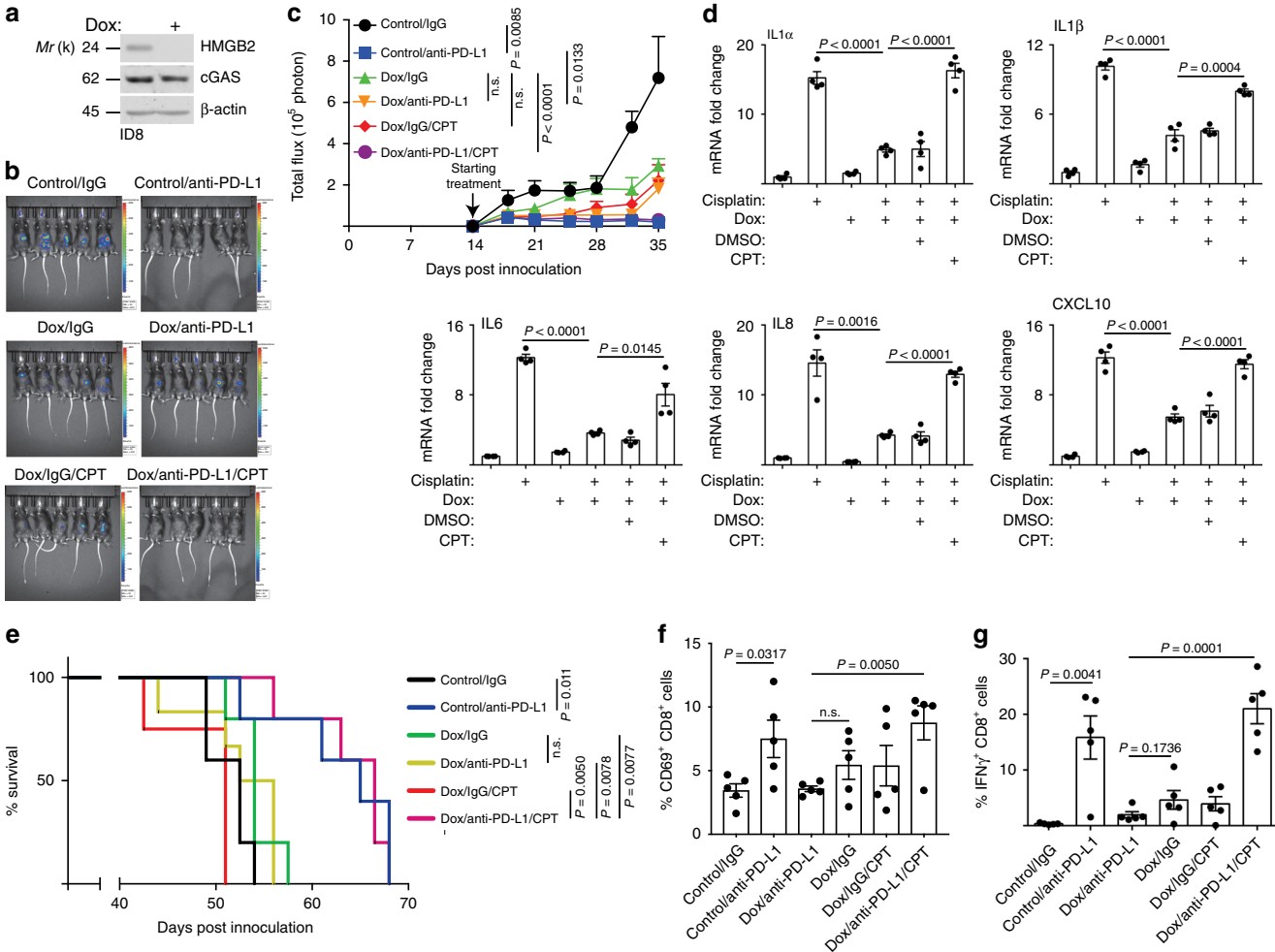

**Fig. 5 HMGB2-TOP1cc-cGAS axis determines response to immune checkpoint blockade. a** Mouse ID8-*Defb29/Vegf-a* ovarian cancer cells expressing doxycycline (DOX) inducible shHMGB2 with or without DOX induction were analyzed for expression of HMGB2, cGAS, and a loading control β-actin by immunoblot. **b** Representative bioluminescence images of mice in the indicated treatment groups at the end of experiments. **c** Quantification of tumor growth based on luciferase bioluminescence in the indicated treatment groups at the indicated time points ($n = 5$ biologically independent mice per group). **d** Expression of the indicated SASP factors in tumor cells sorted by FACS from ascites formed in mice from the indicated groups determined by qRT-PCR ($n = 4$ biologically independent mice per group). **e** After stopping the treatment, the mice from the indicated groups were followed for survival. The figure shows the Kaplan–Meier survival curves ($n = 5$ biologically independent mice per group). **f, g** At the end of treatment, percentage of CD69-positive cells in CD8-positive T cells (**f**) and IFNγ-positive cells in CD8-positive T cells (**g**) was assessed by flow cytometry in the peritoneal wash collected from mice in the indicated treatment groups ($n = 5$ biologically independent mice per group). Data represent mean ± s.e.m. *P*-values were calculated using a two-tailed *t* test except for 4e by log-rank (Mantel–Cox) test. Source data are provided as a Source Data file.

was purchased from Selleck (Cat. No. S4163). Camptothecin was purchased from Selleck (Cat. No. S1288). 4′ 6-Diamidino-2-phenylindole dihydrochloride (DAPI) was purchased from Sigma (Cat. No. D9542). Cytochalasin B was purchased from Sigma (Cat. No. C6762). Spermidine was purchased from Sigma (Cat. No. S2626). Spermine was purchased from Sigma (Cat. No. S3256). Formaldehyde solution was purchased from Sigma (Cat. No. F8775). Paraformaldehyde (PFA) was purchased from Sigma (Cat. No. 158127). The DNA ladder was purchased from Thermo Fisher (Cat. No. SM1333). Benzonase was purchased from Sigma (Cat. No. E1014).

The pMSCVpuro-eGFP-hcGAS, pBABE-puro-H-RAS$^{G12V}$, pBABE-puro-Empty, and pGEX6P1-GST-cGAS plasmids were obtained from Addgene. pLKO.1-shHMGB2 (shHMGB2 #1:TRCN000000150009; shHMGB2 #2: TRCN0000019011) and pLKO.1-shTOP1 (TRCN0000059090) were obtained from the Molecular Screening Facility at the Wistar Institute. pLKO.1-shcGAS short hairpins were purchased from Sigma (TRCN0000146282, TRCN0000149984). pLentiCRISPR-HMGB2 was constructed by inserting the HMGB2 guide RNA (gRNA; 5′-AACA CCCTGGCCTATCCATT-3′) as we previously published[15]. Tet-pLKO-puro-shHMGB2 was constructed using the Tet-pLKO-puro backbone (Addgene, Cat. No. 21915) and shHMGB2 sequence (forward: 5′-CCGGGCTCAACATTAGC TTCAGTATCTCGAGATAC TGAAGCTAATGTTGAGCTTTTTG-3′; reverse: 5′-AATTCAAAAAGCTCAACATTAGCTTCAGTATCTCGA GATACTGAA GCTAATGTTGAGC-3′).

Recombinant cGAS protein was purchased from Cayman (Cat. No. 22810). Recombinant HMGB2 protein was purchased from Prospec (Cat. No. PRO-888).

Recombinant TOP1 protein was purchased from Prospec (Cat. No. ENZ-306). Recombinant TOP1 Y723F mutant protein was purchased from Speed Biosystems (Cat. No. OP10402). Recombinant his-tagged TOP1 protein was purchased from Sino Biological (Cat. No. 17455-H07B). ATP Solution (100 mM) was purchased from Thermo Fisher (Cat. No. R0441). GTP Solution (100 mM) was purchased from Thermo Fisher (Cat. No. R0461). SYBR™ Green I Nucleic Acid Gel Stain was purchased from Thermo Fisher (Cat. No. S7563). (ISD)₂ interferon stimulatory double-strand DNA (dsDNA) was purchased from InvivoGen (Cat. No. tlrl-isdn).

The following antibodies were purchased from the indicated suppliers and used for immunoblotting or immunostaining at the indicated concentrations: mouse monoclonal anti-γH2AX (clone JBW301) (Millipore, Cat. No. 05-636), 1:500 for immunofluorescence; rabbit monoclonal anti-γH2AX (20E3) (Cell Signaling Technology, Cat. No. 9718), 1:500 for immunofluorescence; Alexa Fluor® 594 anti-γH2AX (2F3) (Biolegend, Cat. No. 613410), 1:200 for immunofluorescence; rabbit polyclonal anti-HMGB2 (Abcam, Cat. No. 67282), 1:1000 for immunoblotting and 1:500 for immunofluorescence; mouse monoclonal anti-cGAS (D9) (Santa Cruz, Cat. No. sc-515777), 1:200 for immunofluorescence, 1:1000 for immunoblotting; rabbit monoclonal anti-cGAS (D1D3G) (Cell Signaling Technology, Cat. No. 15102), 1:200 for immunofluorescence, 1:1000 for immunoblotting; rabbit monoclonal anti-STING (D2P2F) (Cell Signaling Technology, Cat. No. 13647 S), 1:1000 for immunoblotting, rabbit polyclonal anti-Cyclin A (H432) (Santa Cruz, Cat. No. sc-751), 1:1000 for immunoblotting; mouse monoclonal anti-RAS (BD Biosciences, Cat. No. 610001), 1:1000 for immunoblotting; mouse monoclonal

anti-P16 (JC8) (Santa Cruz, Cat. No. sc-56330), 1:1000 for immunoblotting; mouse monoclonal anti-P21 (187) (Santa Cruz, Cat. No. sc-817), 1:1000 for immunoblotting; mouse monoclonal anti-β-actin (Sigma, Cat. No. A2228), 1:10,000 for immunoblotting; rabbit polyclonal anti-TOP1 (Proteintech, Cat. No. 20705-1-AP), 1:1000 for immunoblotting and 1:200 for immunofluorescence; mouse monoclonal anti-Topoisomerase I-DNA Covalent Complexes (TOP1cc) (clone 1.1 A) (Millipore, Cat. No. MABE1084), 1:1000 for slot blot and 1:200 for immunofluorescence.

For flow cytometric analysis, APC/CY7 anti-CD69 (Cat. No. 104525), APC anti-CD4 (Cat. No. 100516), PE anti-CD8 (Cat. No. 100708), FITC anti-Granzyme B (Cat. No. 372206), PE/Cy7 anti-interferon-gamma (Cat. No. 505825) antibodies were purchased from Biolegend and used at 1:150 dilutions. Zombie yellow dye (Biolegend, Cat. No. 423103, 1:200) was used as a viability staining.

**Retrovirus and lentivirus infection**. Retrovirus production and transduction were performed using Phoenix cells to package the infection viruses (Dr. Gary Nolan, Stanford University)[26]. Lentivirus was produced using the ViraPower kit (Invitrogen) based on the manufacturer's instructions in the 293FT human embryonal kidney cell line by Lipofectamine 2000 transfection (Thermo Fisher, Cat. No. 11668019). Lentivirus was harvested and filtered with 0.45-μm filter 48 h post transfection. Cells infected with lentiviruses were selected in 1 μg/ml puromycin 48 h post infection.

**Senescence induction and SA-β-Gal staining**. For oncogene-induced senescence, IMR90 cells were infected with retrovirus produced by pBABE-puro-H-RAS^G12V (Addgene) at 37 °C for 24 h. A second round of infection was performed on the same target cells. Infected cells were drug-selected using 3 μg/ml puromycin[26]. For Etoposide-induced senescence, IMR90 or OVCAR3 cells at ~60–70% confluency were treated with 50 μM or 2 μM Etoposide for 48 h. The treated cells were cultured in fresh medium and harvested at day 8. For cisplatin-induced senescence, OVCAR3, or ID8-*Defb29/Vegf-a* cells at ~60–70% confluency were treated with 2 μM cisplatin for 48 h. The treated cells were cultured in fresh medium and harvested at day 8.

SA-β-Gal staining was performed as previously described[26]. Briefly, cells were fixed for 5 min at room temperature in 2% formaldehyde/0.2 glutaraldehyde in PBS. After washing twice with PBS, cells were stained at 37 °C overnight in a non-CO$_2$ incubator in staining solution (40 mM Na$_2$HPO$_4$, pH 6.0, 150 mM NaCl, 2 mM MgCl$_2$, 5 mM K$_3$Fe(CN)$_6$, 5 mM K$_4$Fe(CN)$_6$, and 1 mg/ml X-gal. After counterstaining with Nuclear Fast Red solution (Ricca, Cat. No'. R5463200500), slides were subjected to an alcohol dehydration series and mounted with Permount (Fisher Scientific, Cat. No. SP15-100). Slides were examined using a Zeiss AxioImager A2.

**Secreted cytokine assay**. For cytokine-array analysis, cells were washed once and cultured in serum-free medium for 48 h[26]. Conditioned medium was filtered (0.2 μm) and then subjected to cytokine-array assay using Human Cytokine Array C1 kit (RayBiotech, Cat. No. AAH-CYT-1-2) following the manufacturer's guidelines. After collection of conditional media, the cell number of each sample was counted. The intensities of array dots were visualized on film after incubation with Super-Signal West Pico PLUS Chemiluminescent Substrate (Thermo Fisher Scientific, Cat. No. 34580). The integrated density was measured using Image J, and normalized to the cell number from which the conditioned medium was generated.

**2′ 3′-cGAMP measurement**. 2′ 3′-cGAMP ELISA Kit (Cayman Chemical, Cat. No. 501700) was used to analyze the endogenous level of 2′ 3′-cGAMP following the manufacturer's instructions. In total, $1 \times 10^5$ IMR90 cells were incubated in 200 μL lysis buffer (Thermo Fisher, Cat. No. 78501) on ice for 30 min. The 2′ 3′-cGAMP ELISA was performed following the manufacturer's instructions.

**Immunofluorescence**. Cells were fixed with 4% paraformaldehyde (PFA) for 15 min at room temperature followed by permeabilization with 0.2% Triton X-100 in PBS for 5 min. For DNase I digestion, after fixation, and permeabilization, cells were treated with 500 units/mL RNase-Free DNase I (Qiagen, Cat. No. 79254) for 1 h at 37 °C. After blocking with 1% BSA in PBS, cells were incubated with primary antibody overnight at 4 °C and Alexa-Fluor-conjugated secondary antibody (Life Technologies). Fluorescent images were captured using a Leica TCS SP5 II scanning confocal microscope.

**Immunoblotting and immunoprecipitation**. Cells were lysed in 1× sample buffer (2% SDS, 10% glycerol, 0.01% bromophenol blue, 62.5 mM Tris, pH 6.8, and 0.1 M DTT) and heated to 95 °C for 10 min. Protein concentrations were determined using the protein assay dye (Bio-Rad, Cat. No. #5000006) and Nanodrop. An equal amount of total protein was resolved using SDS-PAGE gels and transferred to PVDF membranes at 110 V for 2 h at 4 °C. Membranes were blocked with 5% nonfat milk in TBS containing 0.1% Tween 20 (TBS-T) for 1 h at room temperature. Membranes were incubated overnight at 4 °C in the primary antibodies in 4% BSA/TBS + 0.025% sodium azide. Membranes were washed four times in TBS-T for 5 min at room temperature, after which they were incubated with HRP-

conjugated secondary antibodies (Cell Signaling Technology, Cat. No. 7076 S, 7074 S) for 1 h at room temperature. After washing four times in TBS-T for 5 min at room temperature, proteins were visualized on film after incubation with Super-Signal West Pico PLUS Chemiluminescent Substrate (Thermo Fisher Scientific, Cat. No. 34580).

For immunoprecipitation, cells were collected and washed once with ice-cold PBS. Whole-cell extracts were lysed with RIPA buffer (50 mM Tris-HCl pH 7.4, 1% NP-40, 150 mM NaCl, 1 mM EDTA, 10% sodium deoxycholate, freshly added with 1 mM phenylmethylsulfonyl fluoride (PMSF), and cOmplete™, EDTA-free Protease Inhibitor Cocktail (Roche, Cat. No. C762Q77)). After $12,000 \times g$ centrifuge for 15 min, the supernatant was collected and incubated with antibody or isotype IgG control (5 μg per sample) at 4 °C overnight, followed by addition of 10 μL of protein A/G-conjugated agarose beads mixture (Thermo Fisher, Cat. No. 10002D and 10004D). The precipitates were washed four times with ice-cold RIPA buffer, resuspended in 2 × Laemmle buffer, and resolved by SDS-PAGE followed by immunoblotting. Unprocessed images of scanned immunoblots shown in Figures and Supplementary Figs. are provided in a Source Data file.

**GST pulldown**. GST pull-down assay was carried out by incubating equal amounts of GST or GST-tagged cGAS (Addgene, Cat. No. 108676) that are immobilized on glutathione-sepharose beads (GE Healthcare, Cat. No. 17-0756-01) with in vitro translated His-tagged TOP1 (Sino Biological, Cat. No. 17455-H07B) at 4 °C for 16 h. Precipitated proteins were washed three times with elution buffer, including 150 mM NaCl, eluted with SDS sample buffer, and subjected to immunoblot analysis.

**Quantification PCR with reverse transcription**. The total RNA was isolated using Trizol (Invitrogen) according to the manufacturer's instruction. Extracted RNAs were used for reverse-transcriptase PCR (RT-PCR) with High-Capacity cDNA Reverse Transcription Kit (Thermo Fisher, Cat. No. 4368814). Quantitative PCR (qPCR) was performed using iTaq™ Universal SYBR® Green Supermix (BIO-RAD, Cat. No. 1725121) and QuantStudio 3 Real-Time PCR System.

The primers sequences used for quantitative RT-PCR are as follows:

Human *IL1α* (forward: 5′-AGGAGAGCCGGGTGACAGTA-3′, reverse: 5′-TC AGAATCTTCCCGTTGCTTG-3′);

Human *IL1β* (forward: 5′-AGCTCGCCAGTGAAATGATGG-3′, reverse: 5′-G TCCTGGAAGGAGCACTTCAT-3′);

Human *IL6* (forward: 5′-ACATCCTCGACGGCATCTCA-3′; reverse: 5′-TCAC CAGGCAAGTCCTCCTCA-3′);

Human *IL8* (forward: 5′-GCTCTGTGTGAAGGTGCAGT-3′; reverse: 5′-TGC ACCCAGTTTTCCTTGGG-3′);

Human *CXCL10* (forward: 5′-CCATTCTGATTTGCTGCCTTATC-3′; reverse: 5′-TACTAATGCTGATGCAGGTACAG-3′);

Human *CCL5* (forward: 5′-CCAGCAGTCGTCTTTGTCAC-3′; reverse: 5′-CT CTGGGTTGGCACACACTT-3′);

Human *ISG15* (forward: 5′-GAGCATCCTGGTGAGGAATAAC-3′; reverse: 5′-CGCTCACTTGCTGCTTCA-3′);

Human *B2M* (forward: 5′-GGCATTCCTGAAGCTGACA-3′; reverse: 5′-CTT CAATGTCGGATGGATGAAAC-3′).

Mouse *IL1α* (forward: 5′-CCAGAAGAAAATGAGGTCGG-3′, reverse: 5′-AG CGCTCAAGGAGAAGACC-3′);

Mouse *IL1β* (forward: 5′-TGTGCAAGTGTCTGAAGCAGC-3′, reverse: 5′-TG GAAGCAGCCCTTCATCTT-3′);

Mouse *IL6* (forward: 5′-GCTACCAAACTGGATATAATCAGGA-3′; reverse: 5′-CCAGGTAGCTATGGTACTCCAGAA-3′);

Mouse *IL8* (forward: 5′-AGAGGCTTTTCATGCTCAACA-3′; reverse: 5′-CCA TGGGTGAAGGCTACTGT-3′);

Mouse *CXCL10* (forward: 5′-TCAGCACCATGAACCCAAG-3′; reverse: 5′-CT ATGGCCCTCATTCTCACTG-3′);

Mouse *B2M* (forward: 5′-AGTTAAGCATGCCAGTATGGCCGA-3′; reverse: 5′-ACATTGCTATTTCTTTCTGCGTGC-3′).

**CCF purification**. CCF purification protocol was developed by modifying previous protocols[27,28]. Briefly, 500 million senescent cells were collected, resuspended, and incubated in DMEM containing 10 μg/mL cytochalasin B for 30 min at 37 °C. After wash once with ice-cold PBS, the cell pellet was gently dounce homogenized in ice-cold pre-chilled lysis buffer (10 mM Tris-HCl, 2 mM magnesium acetate, 3 mM CaCl$_2$, 0.32 M sucrose, 0.1 mM EDTA, 1 mM DTT, 0.1% NP-40, 0.15 mM spermine, 0.75 mM spermidine, 10 μg/ml cytochalasin B, pH 8.5, 4 °C) with ten slow strokes of a loose-fitting pestle. Release of nuclei was confirmed by DAPI staining and microscopy. The homogenate was fixed with 1% formaldehyde for 10 min, and mixed well with an equal volume of 1.6 M sucrose buffer (10 mM Tris-HCl, 5 mM magnesium acetate, 0.1 mM EDTA, 1 mM DTT, 0.3% BSA, 0.15 mM spermine, 0.75 mM spermidine, pH 8.0, 4 °C). A 10 mL portion of homogenate was layered on the top of sucrose buffer gradient (20 mL and 15 mL containing 1.8 M and 1.6 M of sucrose, respectively) in a 50 -mL tissue culture tube. The tubes were centrifuged at $1200 \times g$ for 20 min at 4 °C. After centrifugation, the upper 3 mL of the gradient was discarded, and the next 15 mL containing CCFs was collected. The collected fraction was diluted with an equal volume of ice-cold PBS, and filtered through 5 μm low protein-binding durapore (PVDF) membrane (Millipore, Cat.

No. SLSV025LS) to remove the contaminated nuclei. DAPI staining was performed at this step to confirm the clearance of contaminated nuclei. The CCF fractions were diluted fivefold by adding ice-cold PBS, then centrifuged at $2000 \times g$ for 15 min at 4 °C. Finally, the pellet was suspended in 200 μL ice-cold PBS buffer. The CCF samples were broken down by one pulse of bioruptor with high output. DNA concentration was measured using Nanodrop, and 5 μg of DNA was used for slot blot analysis.

**SILAC-MS analysis.** SILAC DMEM Lysine (6) Arginine (10) Kit (Silantes, Cat. No. 282986434) was used for the SILAC-MS analysis. Briefly, IMR90 cells were cultured in "heavy" medium containing $^{13}C_6$-labeled lysine and $^{13}C_6$, $^{15}N_4$-labeled arginine, or "light" medium containing unlabeled lysine and arginine for at least four passages. The "heavy" labeled IMR90 cells were infected with short hairpin control lentivirus, and the "light" cells were infected with shHMGB2 short hairpin lentivirus (TRCN0000019011). After puromycin selection, the cells were treated with 50 μM Etoposide for 2 days. After washing off the drug with fresh medium, the treated cells were cultured for 6 days to induce senescence. The same numbers of both "heavy" and "light" labeled cells were mixed together and the CCF purification was performed. Purified CCF were mixed with 5× SDS sample buffer and boiled at 95 °C for 15 min.

LC-MS/MS analysis was performed using a Q Exactive HF mass spectrometer (Thermo Fisher Scientific) coupled with a Nano-ACQUITY UPLC system (Waters). Samples were digested with trypsin and tryptic peptides were separated by reversed phase HPLC on a BEH C18 nanocapillary analytical column (75 μm i. d. × 25 cm, 1.7-μm particle size; Waters) using a 240 min gradient formed by solvent A (0.1% formic acid in water) and solvent B (0.1% formic acid in acetonitrile). Eluted peptides were analyzed by the mass spectrometer set to repetitively scan m/z from 400 to 2000 in positive ion mode. The full MS scan was collected at 60,000 resolution followed by data-dependent MS/MS scans at 15,000 resolution on the 20 most abundant ions exceeding a minimum threshold of 10,000. Peptide match was set as preferred, exclude isotope option and charge-state screening were enabled to reject unassigned, and single-charged ions. The sample was analyzed twice (technical replicate). Peptide sequences were identified using MaxQuant 1.6.2.3[29]. MS/MS spectra were searched against a UniProt human protein database (October 2017) and a common contaminants database using full tryptic specificity with up to two missed cleavages, static carboxamidomethylation of Cys, and variable oxidation of Met, and protein N-terminal acetylation. Consensus identification lists were generated with false discovery rates set at 1% for protein and peptide identifications. The DAVID bioinformatics resources 6.8 was used for functional classification analysis. The protein list was further filtered to include only proteins classified as "nucleosome and chromosome related" and identified by at least two razor + unique peptides with a minimum absolute fold change of 1.2 in both replicates.

**Chromatin fragment extraction and transfection.** For chromatin fragments extraction, proliferating IMR90 cells were treated with 5 μM or 50 μM Camptothecin for 30 min to induce low or high levels of TOP1cc, respectively. Cells then were incubated with hypotonic buffer (10 mM Tris, pH 7.4, 30 mM NaCl, 3 mM $MgCl_2$, 0.1% NP-40), supplemented with protease inhibitor cocktail, on ice for 10 min[3]. The cells were then centrifuged at $300 \times g$ for 3 min at 4 °C. The supernatant was carefully removed, and the resulting pellets were incubated with benzonase buffer (50 mM Tris pH 7.5, 300 mM NaCl, 0.5% NP-40, 2.5 mM $MgCl_2$) with protease inhibitor cocktail, supplemented with 10 U of benzonase (Sigma, Cat. No. E1014), on ice, for 30 min. The product was centrifuged again at $300 \times g$ for 3 min at 4 °C, and benzonase was inactivated by addition of 15 mM EDTA. The resulting supernatant contains chromatin fragments and soluble nuclear proteins. For the negative controls, buffer without benzonase was used, and the resulting supernatant only contains soluble nuclear proteins without chromatin fragments. The product was then diluted five times with PBS. Slot blot were performed to confirm the TOP1cc level. The chromatin fragments or negative controls were transfected into proliferating IMR90 cells using lipofectamine 2000. Successful transfection was confirmed by immunofluorescence with DAPI staining. Transfected cells were harvested 4 days post transfection, and were used for RT-qPCR or immunofluorescence analysis.

**TOP1 ICE (in vivo complex of enzyme) assay.** Human Topoisomerase 1 ICE Assay Kit (TopoGEN, Cat. No. TG1020-1) was used to isolate protein–DNA samples which contain TOP1–DNA covalent complex (TOP1cc) for slot blot analysis. The isolation was performed following the manufacturer's guidelines. In total, $5 \times 10^5$ cells were used for ICE assay and TOP1cc analysis. Purified CCF samples were sonicated and used for slot blot directly. Briefly, cells were lysed with 300 μL of room-temperature buffer A, and then 115 μL buffer B was added to precipitate DNA. After washing with buffer C, DNA was dissolved in buffer D and buffer E. The DNA samples were kept at 37 °C to promote the recovery. Nanodrop was used to measure the DNA concentration. In all, 5 μg of DNA was used for each slot blot analysis. Bio-Dot SF Microfiltration Apparatus (Bio-Rad, Cat. No. 1706542) was used for slot blot. Quantification was performed using NIH Image J software.

**Electrophoretic mobility shift (EMSA) assays.** EMSA was performed as previously described[20,30]. Briefly, recombinant cGAS was incubated, in the presence or absence of recombinant TOP1 or TOP1 Y723F mutant, with $(ISD)_2$ dsDNA in the cGAMP synthesis buffer at 37 °C for 30 min. The mixtures were loaded on 1% agarose gel using an electrophoresis buffer (40 mM Tris-HCl at pH 10.5). The gels were then stained with SYBR™ Green I Nucleic Acid Gel Stain, and images were acquired using UV Transilluminator (Analytik Jena).

**In vivo orthotopic syngeneic mouse model.** The protocols were approved by the Institutional Animal Care and Use Committee of the Wistar Institute. Results from in vitro experiments were used to determine the in vivo sample size. For orthotopic syngeneic model, luciferase expressing ID8-*Defb29/Vegf-a* with inducible shHMGB2 was pretreated with 2 μM Cisplatin for 48 h to induce CCFs. In total, $5 \times 10^6$ cells were i.p. injected into the peritoneal cavity of C57BL/6 mouse (female, 6–8 weeks old, CRL/NCI)[21]. Animals were randomly assigned to different treatment groups (ten mice/group). The mice in control groups were fed with control rodent diet (Fisher Scientific, Cat. No. 14-726-309). For the HMGB2 knockdown groups, mice were fed with Bio-Serv™ Doxycycline Grain-Based Rodent Diet (Fisher Scientific, Cat. No. 14-727-450) to induce HMGB2 knockdown. Tumor progression was monitored twice a week using a Xenogen IVIS Spectrum in vivo bioluminescence imaging system. Images were analyzed using Live Imaging 4.0 software. Tumor-bearing mice were treated by i.p. injection with isotype control IgG or anti-PD-L1 antibody (Bio X Cell, Cat. No. B7-H1, clone 10 F.9G2, 10 mg kg$^{-1}$) every 3 days with or without simultaneous TOP1 inhibitor camptothecin treatment (Selleck, Cat. No. S1288; 8 mg kg$^{-1}$). For survival analysis, the Wistar Institute IACUC guideline was followed in determining the time for ending the survival experiments (tumor burden exceeds 10% of body weight).

For peritoneal wash, the peritoneal cavity of mice was washed three times with 5 ml PBS. Single-cell suspensions were prepared, and red blood cells were lysed using ACK Lysis Buffer (Thermo Fisher, Cat. No. A1049201). Live/dead cell discrimination was performed using Zombie Yellow™ Fixable Viability Kit (Biolegend, Cat. No. 423104). Cell surface staining was done for 30 min at 4 °C using antibodies against CD3ε (Biolegend, Cat. No. 423104), CD69 (Biolegend, 104525), CD8 (Biolegend, Cat. No. 100708), CD4 (Biolegend, Cat. No: 100516), Granzyme B (Biolegend, Cat. No. 372206), and Interferon gamma (Biolegend, Cat. No. 505825). Intracellular staining was done using an eBioscience fixation/permeabilization kit (Thermo Fisher, Cat. No. 88-8824-00). All data acquisition was done using an LSR II (BD) or FACSCalibur (BD), and analyzed using FlowJo software (TreeStar) or the FlowCore package in the R language and environment for statistical computing.

**Statistical analysis and reproducibility.** Results are representative of a minimum of three independent experiments. All statistical analyses were conducted using GraphPad Prism 6 (GraphPad). The Student's *t* test was performed to determine *P*-values of the raw data unless otherwise stated, where $P < 0.05$ was considered significant. Animal experiments were randomized. There was no exclusion from the experiments.

**Reporting summary.** Further information on research design is available in the Nature Research Reporting Summary linked to this article.

## Data availability

The mass spectrometry proteomics data have been deposited into the MassIVE data repository with the accession number MSV000084789. All the data supporting the findings of this study are available within the article and its supplementary information files and from the corresponding author upon reasonable request. A reporting summary for this article is available as a Supplementary Information file. The source data underlying Figs. 1b, d, f, g, h, i, 2a, c, d, e, f, g, h, i, k, 3a, b, c, d, 4a, b, c, d, 5a, c, d, e, f, g and Supplementary Figs. 1a, c, e, 2a, c, d, e, g, h, 3b, e, 4c, d, e, 5a, b, c, d, f, g, h, i, j, 6a, d, e, 7b, e, f, g, h, i, j, 8a, b, d, e are provided as a Source Data file.

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

## Acknowledgements

We thank Drs. Shelley Berger and Ronen Marmorstein for critical comments and discussion. This work was supported by US National Institutes of Health grants (R01CA160331, R01CA163377, R01CA202919, and R01CA239128 to R.Z., P01AG031862 to R.Z. and P50CA228991 to R.Z.), US Department of Defense (OC150446 and OC180109 to R.Z.), The Honorable Tina Brozman Foundation for Ovarian Cancer Research (to R.Z.) and Ovarian Cancer Research Alliance (Collaborative Research Development Grant to R.Z., and Ann and Sol Schreiber Mentored Investigator Award to S.W.). Support of Core Facilities was provided by Cancer Centre Support Grant (CCSG) CA010815 to The Wistar Institute.

## Author contributions

B.Z., P.L., T.F., T.N., N.F., S.W., J.L., K.M.A. and H.-Y.T. performed the experiments and analyzed the data. B.Z. and R.Z. designed the experiments. Q.L. performed statistical analysis. D.W.S. and R.Z. supervised the studies. B.Z., H.-Y.T., D.W.S. and R.Z. wrote the paper. R.Z. conceived the study.

## Competing interests

The authors declare no competing interests.
