## [Peer Review File · Nature Communications]

Reviewers' comments:

Reviewer #1 (Remarks to the Author):

NCOMMS-19-33589-T

"Topoisomerase 1 cleavage complex enables pattern recognition and inflammation during senescence" by Zhao et al.

The regulation of senescence-associated secretory phenotypes (SASP) is intensely interesting as it is associated with various aspects of the physiological roles of cellular senescence in vivo. Recently, it has become apparent that the accumulation of cytoplasmic chromatin fragments (CCF) plays a key role in provoking SASP through activating cGAS-STING, a cytoplasmic DNA sensing pathway. However, the molecular mechanisms underlying how cGAS recognize CCF have remained unclear as yet. In this paper, Zhao and co-workers uncovered that the TOP1cc, a TOP1 covalently modified DNA complex, as a critical mediator of the recognition of CCF by cGAS through direct interaction between TOP1 and cGAS in a dsDNA dependent manner during senescence. In addition, Zhao et al revealed that HMGB2 functions upstream of the TOP1cc-cGAS axis by stabilizing TOP1cc. These findings provide novel mechanistic insights into how cGAS recognize CCF towards provoking SASP in senescent cells.

Overall, the main conclusions of this study are well supported by thorough experimentation and high-quality data. Therefore, I am enthusiastic about this study and I highly recommend this study for publication in Nature Communications after the following points have been satisfactorily addressed.

Critiques:

1) On page 3, lines 60 and 61, the authors stated that CCF is caused by nuclear membrane blebbing. However, because cell-cycle progression through mitosis following DNA double-strand breaks (DSBs) has been shown to cause the formation of micronuclei (Harding et al. Nature 2017; Mackenzie et al., Nature 2017), it is possible that some CCF is caused by micronuclei formation in senescent cells. The authors should also mention this possibility.

2) In Fig.5, it is unclear how much of SASP is actually blocked by HMGB2 knock-down. The authors should examine the SASP factor expression in this experimental setting.

3) Along similar lines, in Fig. 5b and c, Dox treatment (Dox/IgG) reduced the tumor growth as compared to control (Control/ IgG). This is suggesting that the blockade of SASP inhibits tumor growth in mice, and is consistent with the previous notion that SASP promotes tumor development, depending on the biological context (Rodier and Campisi, JCB 2011). The authors should discuss this point.

4) Because SA- β -gal activity is not a robust marker on its own, other senescence markers are also needed in Supplementary Fig.1b.

Reviewer #2 (Remarks to the Author):

In this manuscript by Zhao et al, they have described the molecular mechanism by which cGAS senses cytoplasmic chromatin fragments during therapy induced senescence and subsequent expression of the SASP. They have shown that TOP1cc, stabilised by HMGB2, is necessary for the

recognition of CCF by cGAS and subsequent expression of the SASP. Their data are convincing and experimental procedures are sound. The findings are novel and would be of interest to the readership. I have some suggestions that I think would make the manuscript stronger prior to publication:

Major points:

1. As TOP1 and TOP1cc are central subjects in the manuscript, the paper will benefit from an extended and better introduction of TOP1cc.
2. The authors show cGAS co-localisation with HMGB2/TOP1cc and then show that this correlates with expression of the SASP (either via qPCR or antibody array). In fact, they do not robustly demonstrate cGAS activation. It would be of benefit to determine the activation (or indeed suppression with HMGB2 knockdown) of downstream cGAS signalling pathways (for example STING dimerization) or direct measurements of cGAMP.
3. While the images showing co-localisation of TOP1cc/gH2AX are convincing (Figure 2b) I cannot find any quantification of this, as seen with similar figures elsewhere in the manuscript. Moreover, the manuscript will improve if TOP1cc/cGAS colocalization is shown and quantified in senescence induction, HMGB2 depletion and CPT treatment samples.
4. Figures 2g and 2h quantification are important. However, Mock transfection control (cells transfected with a sample lacking any chromatin) is not a good control for the experiment. Benzonase digested chromatin from CPT non treated cells would be the correct control. Furthermore, an experiment by transfection of Benzonase digested chromatin from IMR90 control, IMR90RAS senescence, IMR90RAS HMGB2 depleted, and IMR90 RAS HMGB2 depleted treated with CPT will be a much elegant and convincing experiment.
5. The data concerning co-localisation of cGAS with CCF is using over expressed GFP tagged cGAS (pMSCVpuro-eGFP-hcGAS). Can the authors demonstrate similar co-localisation with endogenous cGAS?
6. In supplementary figure 3e, the authors only show increased relative expression of the SASP factors IL6/CCL5/CXCL10 which is consistent with IRF3 targets. However, the SASP is NFKB mediated, thus do you also see regulation of expression of the key SASP factors IL1a/IL1b and IL8 as shown in previous figures elsewhere in the manuscript?
7. The authors, based on figure 5a-c, conclude that HMGB2 knockdown significantly abrogated the response to PD-L1 antibody treatment. However I cannot see data in this figure that supports this conclusion. From what I can see there is no significant difference between the control/anti-PD-L1 group and the Dox/anti-PD-L1 group (based on there bioluminescence total photon flux quantification in figure 5c). It may be however that a statistical comparison has been done, however this is not made clear in the figure. That sentence could be also deleted as it is not very relevant.

Minor points:

1. A proofread from and native English speaker would be of benefit to correct some minor grammatical errors.

I recommend the publication of the manuscript after some consideration of the points mentioned above.

Reviewer #3 (Remarks to the Author):

The study by Zhao and colleagues focused on delineation of the mechanism of recognition of cytosolic DNA by cGAS. The authors demonstrate that activation of cGAS-dependent SASP signature is dependent on HMGB2-mediated TOP1cc stabilization and localization to cytoplasmic chromatin. They further confirm the importance of this mechanism for response to PD-L1 blockade

in an in vivo model.

Overall, this is a fantastic study that contributes greatly to our understanding of the mechanisms by which cytosolic DNA is recognized by the cGAS pathway. I have several comments that will require clarification by the authors, including some directly dealing with the mechanisms proposed by the team.

1. The primary output of the cGAS-STING pathway is activation of type I IFN and downstream genes (which has been previously demonstrated to be an important pathway for response to immune checkpoint blockade). Oddly, none of the figures presented by the authors show any assessment of type I IFN pathway and only focus on SASP signatures (which can actually be immunosuppressive). Is there any reason why type I IFN pathway was not assessed in these models?
2. Supp. Figure 3f: Presumably figure legend should say "isotope" rather than "isotype"
3. If TOP1 interaction with DNA is destabilized in the absence of HMGB2, how does TOP1 localize to CCF when HMGB2 is knocked down? Can the protein shuttle to CCF independently of DNA?
4. Figure 4: Was anti-TOP1 or anti-TOP1cc antibody used for immunoprecipitation?
5. Figure 4: the authors suggest that interaction between TOP1 and cGAS is direct and is HMGB2-dependent because knockdown of HMGB2 abolished the interaction between the two proteins in co-IP experiments. However, this interaction could simply be compartment-dependent, as TOP1cc does not localize to CCF in the absence of HMGB2. The interaction between TOP1 and cGAS is thus possibly not direct but through both of the proteins interacting with DNA (which is also supported by the ISD addition and EMSA experiments). I think that in the absence of direct co-precipitation or pulldown data demonstrating cGAS and TOP1 interaction (in absence of DNA), it is hard to make a case that the two proteins can interact directly without addition of DNA.
6. Figure 5F: The percentages of IFN γ + CD8 cells are quite high (close to 50%), which is not typical for these types of experiments. Gating strategy for a representative sample (in supplementary) shows only 7% positivity. How was this experiment performed? Also, the Dox/IgG mice demonstrate high levels of IFN γ + CD8 T cells. Why is that the case? Would we not expect this to be the same as control, or perhaps even lower?

Point-by-Point Response to The Reviewers Comments

We sincerely thank both the Reviewers and the Editors for the constructive and thoughtful review provided for our manuscript. We are grateful for their shared appreciation of our manuscript as “*Overall, the main conclusions of this study are well supported by thorough experimentation and high-quality data (Reviewer 1)*”, “*The findings are novel and would be of interest to the readership (Reviewer 2)*”, and “*Overall, this is a fantastic study that contributes greatly to our understanding of the mechanisms by which cytosolic DNA is recognized by the cGAS pathway (Reviewer 3)*”. All the comments raised are truly valuable to improve the manuscript. Correspondingly, we have strived to provide new experimental answers to their comments. I hope that there is no doubt that we have taken the Reviewers’ and Editors’ comments very seriously. We believe that by addressing the reviewers’ concerns we have produced a more solid and cohesive manuscript. A point-by-point response to the reviewers’ comments is detailed below with original comments italicized. Changes that directly address the reviewers’ concerns were denoted with vertical lines in the right margin in the revised manuscript. We hope the Reviewers and the Editors will find this manuscript to be much improved and suitable for publication.

Reviewer #1 (Remarks to the Author):

NCOMMS-19-33589-T

“Topoisomerase 1 cleavage complex enables pattern recognition and inflammation during senescence” by Zhao et al.

The regulation of senescence-associated secretory phenotypes (SASP) is intensely interesting

as it is associated with various aspects of the physiological roles of cellular senescence in vivo. Recently, it has become apparent that the accumulation of cytoplasmic chromatin fragments (CCF) plays a key role in provoking SASP through activating cGAS-STING, a cytoplasmic DNA sensing pathway. However, the molecular mechanisms underlying how cGAS recognize CCF have remained unclear as yet. In this paper, Zhao and co-workers uncovered that the TOP1cc, a TOP1 covalently modified DNA complex, as a critical mediator of the recognition of CCF by cGAS through direct interaction between TOP1 and cGAS in a dsDNA dependent manner during senescence. In addition, Zhao et al revealed that HMGB2 functions upstream of the TOP1cc-cGAS axis by stabilizing TOP1cc. These findings provide novel mechanistic insights into how cGAS recognize CCF towards provoking SASP in senescent cells.

Overall, the main conclusions of this study are well supported by thorough experimentation and high-quality data. Therefore, I am enthusiastic about this study and I highly recommend this study for publication in Nature Communications after the following points have been satisfactorily addressed.

Critiques:

1) On page 3, lines 60 and 61, the authors stated that CCF is caused by nuclear membrane blebbing. However, because cell-cycle progression through mitosis following DNA double-strand breaks (DSBs) has been shown to cause the formation of micronuclei (Harding et al. Nature 2017; Mackenzie et al., Nature 2017), it is possible that some CCF is caused by micronuclei formation in senescent cells. The authors should also mention this possibility.

Response: We thank the reviewer for pointing this out. Accordingly, we now included this possibility on page 3, paragraph 2 in the revised manuscript.

2) In Fig.5, it is unclear how much of SASP is actually blocked by HMGB2 knock-down. The authors should examine the SASP factor expression in this experimental setting.

Response: We thank the reviewer for the comments. As requested, we now examined the SASP factor expression in this experimental setting both *in vitro* and *in vivo* as showed in **New Data Figure 5d and Supplementary Fig. 8a.**

3) Along similar lines, in Fig. 5b and c, Dox treatment (Dox/IgG) reduced the tumor growth as compared to control (Control/ IgG). This is suggesting that the blockade of SASP inhibits tumor growth in mice, and is consistent with the previous notion that SASP promotes tumor development, depending on the biological context (Rodier and Campisi, JCB 2011). The authors should discuss this point.

Response: We thank the reviewer for pointing out this important reference. Accordingly, we now discussed this important point on page 12, paragraph 1 as suggested.

4) Because SA- β -gal activity is not a robust marker on its own, other senescence markers are also needed in Supplementary Fig.1b.

Response: We thank the reviewer for the insightful comments. We now included additional markers of senescence such as expression of p16 and p21 as requested in **New Data Supplementary Fig. 2d-e.**

Reviewer #2 (Remarks to the Author):

In this manuscript by Zhao et al, they have described the molecular mechanism by which cGAS senses cytoplasmic chromatin fragments during therapy induced senescence and subsequent expression of the SASP. They have shown that TOP1cc, stabilised by HMGB2, is necessary for

the recognition of CCF by cGAS and subsequent expression of the SASP. Their data are convincing and experimental procedures are sound. The findings are novel and would be of interest to the readership. I have some suggestions that I think would make the manuscript stronger prior to publication:

Major points:

1. As TOP1 and TOP1cc are central subjects in the manuscript, the paper will benefit from an extended and better introduction of TOP1cc.

Response: We thank the reviewer for the suggestion. We now extended the introduction on TOP1cc as suggested on page 4, paragraph 2.

2. The authors show cGAS co-localisation with HMGB2/TOP1cc and then show that this correlates with expression of the SASP (either via qPCR or antibody array). In fact, they do not robustly demonstrate cGAS activation. It would be of benefit to determine the activation (or indeed suppression with HMGB2 knockdown) of downstream cGAS signalling pathways (for example STING dimerization) or direct measurements of cGAMP.

Response: We thank the reviewer for the insightful comments. We now performed the suggested STING dimerization and cGAMP measurement experiments. Our new data show that HMGB2 knockdown suppressed STING dimerization and cGAMP levels, which can be restored with CPT treatment (**New Data Fig. 2f-g**).

3. While the images showing co-localisation of TOP1cc/gH2AX are convincing (Figure 2b) I cannot find any quantification of this, as seen with similar figures elsewhere in the manuscript.

Moreover, the manuscript will improve if TOP1cc/cGAS colocalization is shown and quantified in senescence induction, HMGB2 depletion and CPT treatment samples.

Response: We now quantified the co-localisation of TOP1cc/ γ H2AX as requested in **New Data Fig. 2c**. In addition, we performed the requested experiments and quantified TOP1cc/cGAS colocalization in control, senescence induction, HMGB2 depletion and CPT treatment samples in **New Data Supplementary Fig. 5e-f**.

4. Figures 2g and 2h quantification are important. However, Mock transfection control (cells transfected with a sample lacking any chromatin) is not a good control for the experiment. Benzonase digested chromatin from CPT non treated cells would be the correct control. Furthermore, an experiment by transfection of Benzonase digested chromatin from IMR90 control, IMR90RAS senescence, IMR90RAS HMGB2 depleted, and IMR90 RAS HMGB2

depleted treated with CPT will

be a much elegant and convincing experiment.

Response: We thank the reviewer for the insightful comments. In fact, we did perform the suggested experiments for Benzonase digested chromatin from CPT non treated control cells as

negative controls. As predicted, chromatin fragments from CPT non treated control cells did not affect cGAS/TOP1cc colocalization (**New Data Fig. 2j-k**). We also performed the suggested

experiments using RAS-expressing IMR90 cells under the suggested conditions. However, due to strong induction of TOP1cc in RAS-induced senescent cells, the levels of TOP1cc in chromatin fragments isolated from nuclear chromatin remain high (as oppose to changes in CCFs, e.g., in Fig. 2d-e) (**Figure 1a for Reviewer**). Consistently, transfection of these TOP1cc high chromatin fragments isolated from these cells induced cGAS/TOP1cc colocalization (**Figure 1b for Reviewer**). Thus, these results are consistent with the notion that transfection of TOP1cc-containing chromatin fragments induces cGAS localization in CCF.

5. The data concerning co-localisation of cGAS with CCF is using over expressed GFP tagged cGAS (pMSCVpuro-eGFP-hcGAS). Can the authors demonstrate similar co-localisation with endogenous cGAS?

Response: We thank the reviewer for the suggestion. We now performed the suggested experiments and showed that endogenous cGAS co-localized with TOP1cc in CCF (**New Data Fig. 1c-d**).

6. In supplementary figure 3e, the authors only show increased relative expression of the SASP factors IL6/CCL5/CXCL10 which is consistent with IRF3 targets. However, the SASP is NFKB mediated, thus do you also see regulation of expression of the key SASP factors IL1a/IL1b and IL8 as shown in previous figures elsewhere in the manuscript?

Response: We now performed the suggested experiments for additional key SASP factors such as IL1a/IL1b and IL8 and the results are included in the **New Data Supplementary Fig. 3e**.

7. The authors, based on figure 5a-c, conclude that HMGB2 knockdown significantly abrogated

the response to PD-L1 antibody treatment. However I cannot see data in this figure that supports this conclusion. From what I can see there is no significant difference between the control/anti-PD-L1 group and the Dox/anti-PD-L1 group (based on there bioluminescence total photon flux quantification in figure 5c). It may be however that a statistical comparison has been done, however this is not made clear in the figure. That sentence could be also deleted as it is not very relevant.

Response: We thank the reviewer for pointing this out. As suggested, we now added statistical analysis to justify the description ($P=0.0133$) on **Fig. 5c**.

Minor points:

1. A proofread from and native English speaker would be of benefit to correct some minor grammatical errors.

Response: The manuscript is now proofread by a native English speaker to correct grammatical errors as suggested.

I recommend the publication of the manuscript after some consideration of the points mentioned above.

Response: We thank the reviewer for the positive comments.

Reviewer #3 (Remarks to the Author):

The study by Zhao and colleagues focused on delineation of the mechanism of recognition of cytosolic DNA by cGAS. The authors demonstrate that activation of cGAS-dependent SASP signature is dependent on HMGB2-mediated TOP1cc stabilization and localization to cytoplasmic chromatin. They further confirm the importance of this mechanism for response to PD-L1 blockade in an in vivo model.

Overall, this is a fantastic study that contributes greatly to our understanding of the mechanisms by which cytosolic DNA is recognized by the cGAS pathway. I have several comments that will require clarification by the authors, including some directly dealing with the mechanisms proposed by the team.

1. The primary output of the cGAS-STING pathway is activation of type I IFN and downstream genes (which has been previously demonstrated to be an important pathway for response to immune checkpoint blockade). Oddly, none of the figures presented by the authors show any assessment of type I IFN pathway and only focus on SASP signatures (which can actually be immunosuppressive). Is there any reason why type I IFN pathway was not assessed in these models?

Response: We thank the reviewer for the insightful comments. This is because we primarily focused on senescence-associated phenotypes and SASP genes such as *CXCL10* and *IL6* examined in the manuscript are indeed targets genes of type I interferon response/IRF3^{1,2}. Regardless, we now examined the effects of the HMGB2 knockdown and CPT treatment on expression of type I IFN downstream target genes such as interferon-stimulated gene 15 (*ISG15*) in **New Data Supplementary Fig. 5g**.

2. Supp. Figure 3f: Presumably figure legend should say “isotope” rather than “isotype”

Response: We thank the reviewer for spotting this and this has now been corrected.

3. If TOP1 interaction with DNA is destabilized in the absence of HMGB2, how does TOP1 localize to CCF when HMGB2 is knocked down? Can the protein shuttle to CCF independently of DNA?

Response: We thank the reviewer for the insightful comment. To address this question, we now performed the DNase I treatment experiment. Our results show that TOP1's localization into CCF is DNA independent. These results are now included as **New Data Supplementary Fig. 6b-c.**

4. Figure 4: Was anti-TOP1 or anti-TOP1cc antibody used for immunoprecipitation?

Response: We now clarified the antibody used for immunoprecipitation was anti-cGAS and anti-TOP1 antibody was used for immunoblotting. Although we tried several times, however, we were unable to perform immunoprecipitation using either anti-TOP1 or anti-TOP1cc antibody.

5. Figure 4: the authors suggest that interaction between TOP1 and cGAS is direct and is HMGB2-dependent because knockdown of HMGB2 abolished the interaction between the two proteins in co-IP experiments. However, this interaction could simply be compartment-dependent, as TOP1cc does not localize to CCF in the absence of HMGB2. The interaction between TOP1 and cGAS is thus possibly not direct but through both of the proteins interacting with DNA (which is also supported by the ISD addition and EMSA experiments). I think that in the absence of direct co-precipitation or pulldown data demonstrating cGAS and TOP1

interaction (in absence of DNA), it is hard to make a case that the two proteins can interact directly without addition of DNA.

Response: We now performed the suggested experiments using purified cGAS and TOP1 in a GST pull-down experiment and our results show the direct interaction between purified cGAS and TOP1 (**New Data Supplementary Fig. 6a**)

6. Figure 5F: The percentages of IFN γ + CD8 cells are quite high (close to 50%), which is not typical for these types of experiments. Gating strategy for a representative sample (in supplementary) shows only 7% positivity. How was this experiment performed? Also, the Dox/IgG mice demonstrate high levels of IFN γ + CD8 T cells. Why is that the case? Would we not expect this to be the same as control, or perhaps even lower?

Response: We are grateful for the reviewer for the comments. We now clarified this point by stating in the figure legend that the percentages of IFN γ + CD8 cells in Figure 5 was calculated against total CD8 positive T cells. In addition, we now re-analyzed the data with more stringent gating-strategy and we can still see a mild increase in the levels of IFN γ + CD8 T cells in Dox/IgG condition, which is consistent with previous reports that expression of shRNA (upon Dox-treatment) can trigger interferon and innate immune response³, which can trigger adaptive immune response to increase IFN γ + CD8 T cells.

Cited References:

- 1 Yanai, H. *et al.* Revisiting the role of IRF3 in inflammation and immunity by conditional and specifically targeted gene ablation in mice. *Proc Natl Acad Sci U S A* **115**, 5253-5258, doi:10.1073/pnas.1803936115 (2018).
- 2 Indraccolo, S. *et al.* Identification of genes selectively regulated by IFNs in endothelial cells. *J Immunol* **178**, 1122-1135, doi:10.4049/jimmunol.178.2.1122 (2007).
- 3 Bridge, A. J., Pebernard, S., Ducraux, A., Nicoulaz, A. L. & Iggo, R. Induction of an interferon response by RNAi vectors in mammalian cells. *Nat Genet* **34**, 263-264, doi:10.1038/ng1173 (2003).

REVIEWERS' COMMENTS:

Reviewer #1 (Remarks to the Author):

I found the revised manuscript has been improved and indeed addressed to critiques raised by me. Therefore, I think this manuscript is now suitable for publication in Nature Communications.

Reviewer #2 (Remarks to the Author):

The paper has improved substantially, the authors have answered all the concerns. The paper is relevant and provides important mechanistic insight into cGAS-STING regulation upon genotoxic stress. I strongly recommend the publication of the manuscript.

Reviewer #3 (Remarks to the Author):

I commend the authors for rapidly addressing the comments through additional experimentation. I have no further comments.

Point-by-Point Response to The Reviewers Comments

REVIEWERS' COMMENTS:

Reviewer #1 (Remarks to the Author):

I found the revised manuscript has been improved and indeed addressed to critiques raised by me. Therefore, I think this manuscript is now suitable for publication in Nature Communications.

Response: We thank the reviewer for the positive comments.

Reviewer #2 (Remarks to the Author):

The paper has improved substantially, the authors have answered all the concerns. The paper is relevant and provides important mechanistic insight into cGAS-STING regulation upon genotoxic stress. I strongly recommend the publication of the manuscript.

Response: We thank the reviewer for the positive comments.

Reviewer #3 (Remarks to the Author):

I commend the authors for rapidly addressing the comments through additional experimentation. I have no further comments.

Response: We thank the reviewer for the positive comments.